# Long-Term Survival and Quality of Life in Non-Surgical Adult Patients Supported with Veno-Arterial Extracorporeal Oxygenation

**DOI:** 10.3390/jcm11216452

**Published:** 2022-10-31

**Authors:** Tomaž Cankar, Mihela Krepek, Marinos Kosmopoulos, Peter Radšel, Demetris Yannopoulos, Marko Noc, Tomaž Goslar

**Affiliations:** 1Department of Intensive Internal Medicine, University Medical Centre Ljubljana, 1000 Ljubljana, Slovenia; 2Medical Faculty, University of Ljubljana, 1000 Ljubljana, Slovenia; 3Division of Cardiology, Department of Medicine and Center for Resuscitation Medicine, University of Minnesota Medical School, Minneapolis, MN 55455, USA

**Keywords:** health-related quality of life, survival, cardiac arrest, cardiogenic shock, veno-arterial extracorporeal membrane oxygenation

## Abstract

Background: The use of veno-arterial extracorporeal membrane oxygenation (VA ECMO) for hemodynamic support is on the rise. Not much is known about the impact of extracorporeal membrane oxygenation (ECMO) and its complications on long-term survival and quality of life. Methods: In this single-center, cross-sectional study, we evaluated the survival and quality of life in patients treated with VA ECMO between May 2009 and July 2019. Follow-up was conducted between November 2019 and January 2020. Results: Overall, 118 patients were evaluated in this study. Of the 37 patients who were alive at hospital discharge, 32 answered the EuroQol-5 dimensional—5-level questionnaire (EQ-5D-5L). For patients discharged alive from the hospital, mean survival was 8.1 years, 8.4 years for cardiogenic shock, and 5.0 years for patients with refractory cardiac arrest. EQ-5D-5L index value of ECMO survivors was not significantly different from the general age-matched population. Neurologic complications and major bleeding during index hospitalization limit long-term quality of life. Conclusions: Patients treated with VA ECMO have high in-hospital mortality, with extracorporeal membrane oxygenation cardio-pulmonary resuscitation patients being at higher risk of early death. However, once discharged from the hospital, most patients remain alive with a reasonable quality of life.

## 1. Introduction

Veno-arterial extracorporeal membrane oxygenation (VA ECMO) is increasingly used for mechanical circulatory support of patients with either cardiogenic shock (CS) or refractory cardiac arrest (RCA) treated with extracorporeal membrane oxygenation cardiopulmonary resuscitation (eCPR) [1,2]. Cardiogenic shock is defined as inability of the heart to maintain adequate cardiac output in accordance with metabolic demands attributed to predominantly cardiac pathology. When established treatment with fluid status optimization, vasopressors and inotropes, mechanical ventilation, and etiologic treatment of cardiogenic shock, whenever possible, fails to restore adequate cardiac output, temporary mechanical circulatory support with VA ECMO is an option to restore adequate organ perfusion [3]. Randomized trials are underway to investigate whether mechanical support by extracorporeal membrane oxygenation (ECMO) provides a survival benefit in cardiogenic shock. Cardiac arrest is an extreme form of cardiogenic shock, and when conventional cardiopulmonary resuscitation fails to restore sustained heart function, cannulation while performing chest compressions and initiating VA ECMO, termed eCPR, is often the only hope for survival [4]. We have the first randomized evidence that it is indeed superior to conventional cardiopulmonary resuscitation for cardiac arrest [5]. Currently, the reported survival rate at hospital discharge for patients with cardiogenic shock supported with VA ECMO is 41–64% [6,7], whereas for refractory cardiac arrest patients treated with eCPR, it is highly dependent on the site of cardiac arrest (in-hospital or out-of-hospital) [5,8,9] and the site of cannulation (cath lab or out-of-hospital) [5,8]. Currently, the survival rate for RCA treated with eCPR ranges from 8–54% [1,5,8,10].

There is a body of evidence supporting the short-term survival benefit of VA ECMO for CS and RCA [11,12,13,14]. However, little is known about long-term survival beyond the first year post discharge and the health-related quality of life (HRQoL) in discharged patients.

Cardiac arrest is an extreme form of CS, and it is often difficult to distinguish between CS being cannulated for VA ECMO and RCA patients who require eCPR. Because of the different baseline characteristics according to treatment strategy [15], these two groups were evaluated separately.

Patients treated for CS or RCA with VA ECMO belong to a very sick population with high in-hospital mortality and complication rates due to both the severity of the underlying disease and the ECMO treatment. Many of these problems may persist after discharge and affect post-hospital recovery and quality of life.

The EuroQol-5 dimensional (EQ-5D) and the short-form health survey with 36 questions (SF-36) are the most commonly used questionnaires to assess health-related quality of life in clinical trials. The EQ-5D questionnaire exists in two forms: 3-level (EQ-5D-3L) and 5-level (EQ-5D-5L). The EQ-5D-5L is a newer version, available since 2011, that provides more precise and valid results and is recommended as the assessment tool of choice [16]; therefore, it was used in our study.

This report aims to assess the long-term survival of patients treated with VA ECMO and their HRQoL, to compare differences in long-term survival and HRQoL between patients with cardiogenic shock and patients with refractory cardiac arrest, and to investigate their possible influences on HRQoL.

## 2. Materials and Methods

### 2.1. Study Population

The study was approved by the Republic of Slovenia national medical ethics committee. All patients treated at the Department of Intensive Internal Medicine at the University Medical Centre Ljubljana, Ljubljana, Slovenia, with VA ECMO from May 2010 to July 2019 were included. Patients were identified from our prospective registry of ECMO patients. Treatment with VA ECMO for either refractory cardiogenic shock and refractory in-hospital or out-of-hospital cardiac arrest was the main inclusion criterion. Patients treated with veno-venous, veno-arterio-venous or crossover to veno-venous ECMO were excluded from follow-up and analysis. Missing clinical data were retrieved from patients’ medical records.

Refractory cardiac arrest patients were considered for eCPR if they met the inclusion/exclusion criteria followed by our center, which are: age younger than 60 years, witnessed cardiac arrest (out-of-hospital or in-hospital), immediate bystander cardiopulmonary resuscitation, initial shockable rhythm (presumed to be of cardiac origin), expected time from cardiac arrest to initiation of ECMO of less than 60 min, no known comorbidities (end-stage renal disease, end-stage liver disease, dementia, or severe neurologic deficit), and no do not resuscitate order or resident of a nursing home.

Initiation of mechanical support in refractory CS patients was at discretion of the attending physician when other means of treatment, including vasopressors and inotropes, mechanical ventilation, and occasionally an intra-aortic balloon pump, were deemed inadequate.

The inclusion and exclusion criteria remained the same throughout the study period. However, at baseline, some patients with prolonged cardiac arrest were resuscitated by eCPR, and in recent years, because of poor outcomes, we have begun to discontinue treating patients who have cardiogenic shock with mechanical complications of myocardial infarction with VA ECMO.

Patients were classified into six (6) subgroups according to the cause of CS and RCA: acute coronary syndrome, myocarditis, dilated cardiomyopathy (DCMP), pulmonary embolism, valvular heart disease, and other causes, which included primary graft rejection, aortic dissection, toxic cardiomyopathy, uncontrollable arrythmias, hypothermia, poisoning, and infection.

Patients were considered to have serious preexisting health problems if they had New York Heart Association Heart stage heart failure III or IV, a history of stroke, an organ transplant, or had organ failure requiring an organ transplant.

To assess the severity of the disease, we calculated the Sequential Organ Failure Assessment (SOFA) [17] and Acute Physiology And Health Evaluation II (APACHE II) [18] scores for the patients who survived in the first 12 h after ECMO initiation. The worst physiological or laboratory values 12 h after cannulation were used for the calculations. We also calculated the SAVE score based on the worst values in the 6 h before ECMO [19]. For RCA patients, pulse pressure below 20 mmHg, diastolic pressure below 40 mmHg, cardiac arrest before ECMO, and central nervous system dysfunction were used for each case.

CardioHelp (Maquet—Getinge group, Rastatt, Germany) or Levitronix CentriMag (Levitronix LLC, Waltham, MA, USA) were used for ECMO support. Cannulation was performed either percutaneously under fluoroscopic guidance by the attending intensive care unit (ICU) physician or surgically with prior direct vascular visualization by the attending cardiovascular surgeon. Patients were heparinized to maintain an activated partial thromboplastin time above 50 s unless active bleeding occurred. ECMO flows were kept as low as necessary to maintain organ perfusion, with 1.5 L/min being the minimum cut-off point. Decannulation was performed by cardiovascular surgeons with vascular suturing.

Complications during ECMO treatment were defined as: cannulation-related complications, limb ischemia, neurological complications (stroke, intracranial hemorrhage, or proven brain death), sepsis (positive blood cultures or proven source of infection with corresponding increase in laboratory markers of infection (leukocyte count, C-reactive protein, and procalcitonin) and increased need for vasopressors), thromboembolism, and bleeding, the severity of which was classified according to Bleeding Academic Research Consortium (BARC) criteria and dichotomized into clinically unimportant (BARC 0–2) and clinically important (BARC 3–5).

### 2.2. Long Term Survival

All patients who were cannulated and received VA ECMO support were included in the analysis.

For patients discharged alive from the hospital, survival status or time of death was determined from Slovenian national health insurance records between November 2019 and January 2020. For patients who participated in the follow-up with EuroQol-5 dimensional—5-level questionnaire (EQ-5D-5L), the interview date was used as the censoring date. The longest follow-up period was 3474 days (9.5 years).

### 2.3. Quality of Life

We contacted eligible patients via phone call and asked them to complete the EQ-5D-5L questionnaire between November 2019 and January 2020. Participants who could not be contacted by phone call received a written form of the questionnaire and a prepaid envelope.

The EQ-5D-5L is a standardized measure developed by the EuroQol group to provide a simple, generic measure of health for clinical and economic appraisal [20]. It has been well-validated in several studies for critical care populations ([21,22]) but has only been used once in an ECMO population [23]. We are not aware of any studies that have validated EQ-5D in ECMO populations.

The EQ-5D-5L assesses five dimensions of health, namely mobility, self-care, usual activities, pain/discomfort, and anxiety/depression, with five severity levels: no problems, slight problems, moderate problems, severe problems, and unable to perform a task. The visual analog scale (EQ-VAS) is a constituent part of the questionnaire and provides a quantitative measure of the patient’s general perception of health on a scale of 0 to 100 [24].

The EQ-5D-5L health states defined by the EQ-5D-5L descriptive system can be converted into a single health-state index value (also referred to as preference weights or preference-based values). Using the set of weights obtained from population-specific validation studies, one can convert each of the 3691 EQ-5D-5L health states into a single summary index value. The index value reflects how good or bad the health state is according to the preferences of the general population of a country or region, with 1 representing full health and 0 representing a state equivalent to death [20]. The index value is a measure of the health status of the general population of a country or region. Self-perception of one’s well-being is highly subjective and culturally dependent. A person’s perception of health should be made in the context of the social and cultural environment in which they live [25]. Unfortunately, Slovenia does not yet have its own country-specific EQ-5D-5L index value dataset. Golicki et al. have suggested using the Polish index value dataset [26].

### 2.4. Statistical Analysis

Continuous variables were expressed as mean and standard deviation (SD), and categorical data were presented as absolute and relative frequencies. The *t*-test for independent samples assuming unequal variances was used for comparison of continuous variables, and the chi-square test was used for comparison of categorical variables. The nonparametric Mann–Whitney test was used for comparison of non-normally distributed variables. The Pearson correlation was used to assess correlation and the chi-square test for independence with Cramer’s V test to test for association. Survival was calculated using Kaplan–Meier analysis. Differences in survival were compared using the log-rank test. Follow-up time was calculated using the reverse Kaplan–Meier method and expressed as median and interquartile range (IQR). Multivariate Cox regression analysis was then performed using as covariates the predictors of mortality, including age, sex, arterial hypertension, diabetes, dyslipidemia, smoking, coronary artery disease, peripheral artery disease, BMI, and shock etiology. A significance level of *p* < 0.05 was used. All statistical analyses were performed using SPSS 22.0 (IBM, SPSS Inc., Chicago, IL, USA).

## 3. Results

From May 2010 to October 2019, 118 patients were treated with VA ECMO mechanical support, 72 (61%) for CS management, and 46 (39%) for RCA. Forty-four patients (37.3%) treated with VA ECMO were discharged alive from the hospital, thirty-one (43.1%) after CS, and thirteen (28.3%) after RCA. Demographic, clinical, and severity of illness data are shown in Table A1.

The survival status of two (1.7%) patients who were foreign residents could not be assessed at the time of follow-up; therefore, they were censored in the survival analysis at the time of hospital discharge. Of the remaining 116 patients whose survival status could be assessed at the time of follow-up, 37 (31.9%) were still alive: 26 (36.1%) in the CS group, and 11 (23.9%) in the RCA group.

Acute coronary syndrome occurred significantly more often in the RCA group than in the CS group. RCA patients also had significantly higher baseline lactate, SOFA, APACHE II, and SAVE scores, while CS patients were more often additionally supported with IABP and had more severe pre-existing health problems, longer duration of ECMO support, a longer ICU stay, and a longer hospital stay (Table A1).

The median follow-up time was 969 days (IQR 500–2192) (2.7, IQR 1.4–6.0 years) for the patients studied, 1128 days (IQR 500–2355) (3.1, IQR 2.7–6.5 years) for the CS patients, and 752 days (IQR 369–1122) (2.1, IQR 1.01–3.1 years) for the RCA patients. (Figure 1) For patients discharged alive from the hospital, mean survival was 2943 days (95% CI: 2521–3364) (8.1, 95% CI 6.9–9.2 years), 3060 days (95% CI: 2636–3485) (8.4, 7.2–9.5 years) for the CS patients, and 1817 days (95% CI: 1256–2378) (5.0, 95% CI 3.4–6.5 years) for the RCA patients. The difference in survival from hospital discharge was not statistically significant (log-rank *p* = 0.282) (Figure 2). The difference in survival persisted even after adjusting the survival curves for possible cofounders (Figure A1, Table A3, Figure A2, Table A4). Mortality in the first 30 days after ECMO implantation was high: 56.8% overall, 51.4% for CS, and 65.2% for RCA patients.

At the time of follow-up, 33 (89.2%) of the patients still alive answered the EQ-5D-5L questionnaire: 24 from the CS group and 8 from the RCA group. Their baseline characteristics are summarized in Table 1. The results of EQ-5D-5L are shown in Table 2. The health dimension where the fewest participants reported problems was self-care, where 20 (62.5%) reported no problems. However, two patients (6.3%) reported extreme problems. Two patients (6.3%) also reported extreme problems in the usual activities dimension (Figure 3). There were no statistically significant difference between CS and RCA patients in any of the EQ-5D-5L attributes although RCA survivors reported numerically lower EQ-VAS values and higher EQ-5D-5L index values compared with CS survivors.

Of the 37 patients who were alive at the time of follow-up, 12 (32.4%) were employed: 8 (30.7%) in the CS group and 4 (36.4%) in the RCA group. The remainder were either retired or eligible for disability benefits.

Patients with cannulation complications, limb ischemia, sepsis, and clinically significant bleeding had higher 30-day mortality, whereas neurologic and thrombotic complications did not affect 30-day survival (Table A2).

There was no significant association between the etiology of hemodynamic deterioration (acute coronary syndrome, myocarditis, valvular heart disease, dilated cardiomyopathy, pulmonary embolism, other) requiring VA ECMO initiation and any measure of health status (problem severity for any of the five health dimensions, EQ-5D-5L index value, or EQ-VAS). However, there was a strong association between neurological complications and self-care (*p* = 0.026; V = 0.534), pain/discomfort (*p* = 0.040, V = 0.509), and anxiety/depression (*p* = 0.001, V = 0.758). There was no association between neurological complications and the other health dimensions, EQ-5D-5L index values, or EQ-VAS. No association was observed between the other complications assessed (defined as clinically significant bleeding), thrombotic events, cannulation-related complications, limb ischemia or sepsis), and the HRQoL measures.

No significant association was observed between length of follow-up and either EQ-5D-5L index value or EQ-VAS (r (32) = −0.14, *p* = 0.225 and r (32) = 0.23, *p* = 0.108, respectively). The distributions of EQ-5D-5L index values and EQ-VAS over time are shown in Figure 4.

## 4. Discussion

To our knowledge, this is the study with the longest follow-up of non-surgical VA ECMO patients focusing primarily on long-term HRQoL assessed with the EQ-5D-5L questionnaire. We describe our 10-year, single-center experience with VA ECMO use in CS and RCA patients. Despite the increasing use of mechanical circulatory support in recent years, early in-hospital mortality remains high [6]. Therefore, the evaluation of long-term outcomes is necessary to investigate whether this expensive and labor-intensive treatment option is justified. Consistent with previous reports, our results suggest that patients treated with VA ECMO have low post-hospital discharge mortality and relatively good HRQoL. However, direct comparison with other studies is difficult due to differences in the quality-of-life scales used.

A number of studies have assessed HRQoL after ECMO using either the EQ-5D-3L or the SF-36 [7,10,19,23,25,28,29,30,31,32,33,34]. Their results can be summarized as follows. Follow-up of HRQoL ranged from 11 months to 9 years. A trend of improvement in HRQoL was observed over the first 12 months after discharge. HRQoL scores in ECMO patients tended to be lower than in the reference population without ECMO but higher than in other acute conditions such as myocardial infarction, heart failure, hemodialysis, and cardiothoracic surgery [35].

To our knowledge, few studies have used the EQ-5D-5L questionnaire to assess VA ECMO patients. The EQ-5D-5L questionnaire was also used by Rolle and colleagues for HRQoL assessment but included a mixed ECMO population receiving ECMO for either circulatory support (VA ECMO *n* = 14) or respiratory support (veno-venous ECMO *n* = 19) [23]. Comparing them to our own cohort in terms of mean dimension score, only self-care was significantly worse in our group (1.7 ± 1.1, *n* = 32 vs. 1.2 ± 0.8, *n* = 33; *p* = 0.040). The difference in EQ-VAS scores was not significant. Since 58% of Rolle’s group had ECMO for lung support, the comparison may not be the most appropriate.

Survival after serious illness is only one aspect of the follow-up. How well survivors function in their lives and how they perceive their well-being in the physical, mental, and social aspects of their lives are just as important as survival. When we compare ECMO survivors aged 54.1 ± 8.8 years with the general Polish population, there is no difference in EQ-5D-5L index value (mean ± SD: 0.864 ± 0.191, *n* = 32) with either the younger age group (45 to 54 years; mean ± SD: 0.898 ± 0.148, *n* = 612; *p* = 0.213) or with the older age group (55 to 64 years; mean ± SD: 0.856 ± 0.141, *n* = 797; *p* = 0.757) [27]. It can be concluded that VA ECMO patients not only have low mortality after hospital discharge but also comparable self-perceived health status. Their life satisfaction indices are also comparable to those of an age-matched population. The high early mortality and stable long-term survival are consistent with the previously published study by Harley et al., with a reported mortality of less than 10% between 1 and 5 years of follow-up [36].

Hellevuo et al. [37] indicated that HRQoL remains good in survivors 6 months after cardiac arrest and that it is mainly influenced by HRQoL before cardiac arrest. To some extent, this might be true for VA ECMO patients. RCA patients in our cohort were mostly healthy before cardiac arrest, in contrast to CS patients who often had severe preexisting health problems. The same trend was seen in EQ-5D-5L index values, which were numerically higher in RCA patients and had lower numerical EQ-VAS scores compared to CS patients. EQ-VAS represents a more personal perception of health, whereas index values are used to represent socially perceived health status. EQ-VAS, which has been reported in other published series, was quite similar to ours [38]. A higher dispersion is observed on EQ-VAS, especially in the RCA group. There could be several explanations for the higher dispersion. The EQ-VAS also has greater dispersion in the general population [39]. In contrast to the EQ-5D-5L index values, which decrease over time, we observed a trend toward improvement in EQ-VAS in our population. Because patients in the RCA group had a shorter follow-up period, one might expect an improvement in personal health perception over time and also a smaller dispersion of EQ-VAS.

Previous studies suggest a significant time-dependent improvement in SF-36 scores in the first year after treatment with ECMO [23,34]. We did not observe significant changes in HRQoL measures over time. The median follow-up time was more than 2 years after ECMO. Therefore, it is plausible that the improvement in HRQoL measures might stop at a certain point. A larger cohort of patients or repeated HRQoL assessments would be needed to confirm this.

Looking at the health dimensions individually, the fewest problems were reported in self-care, as 20 patients (62.5%) reported no problems. On the other hand, most impairments were in self-care and usual activities, where two patients (6.3%) reported extreme problems or were unable to perform tasks. All of these extreme problems were reported in the CS subgroup of patients. Thus, we conclude that although most patients are coping well and report no serious health problems, a minority is still severely disabled and has major limitations in their daily lives.

Anxiety and depression was the health dimension where the majority of patients reported at least some problems; only 12 (37.5%) patients reported no anxiety and depression. The prevalence of anxiety in our study is consistent with previous reports [23]. Anxiety is commonly reported in ICU survivors. In fact, clinically significant anxiety was reported in up to 45.4% of ICU patients 6 weeks after discharge [40]. Whether the high levels of anxiety can be explained by the ICU stay alone or are related to ECMO remains unknown.

CS patients have significantly better overall survival compared to their peers who required ECMO for RCA (*p* = 0.008) (Figure 1; Figure A1 and Figure A2; Table A3 and Table A4). However, this is mainly attributed to higher in hospital mortality in the RCA group. There was no significant difference between survival of CS and RCA patients after hospital discharge (*p* = 0.282) (Figure 2). There were fewer patients in the RCA group who required rehospitalization after discharge from the index hospitalization. However, the rate of rehospitalization corrected per patient-years was similar in both groups.

The main limitation of our study is the retrospective collection of some baseline data. However, data collection for the primary endpoints of the study was done prospectively. The collection of questionnaires at different follow-up time points is a known shortcoming of cross-sectional design. In addition, this is a single-center cohort of patients treated over a 10-year period. Treatment strategies may have changed slightly over this period, and survival may have improved with increased experience with ECMO. Selection bias may be present because the decision to initiate ECMO treatment was made by different attending physicians and because patients denied ECMO were not studied in this paper. The small number of patients limits the power of the statistical analysis. Some patients refused to participate in the study, and some were lost to follow-up. However, our loss to follow-up and dropout rates was comparable to those of other published studies [16,21]. The use of the EQ-5D-5L questionnaire is another limitation. The EQ-5D-5L has only been validated for the ICU population and not specifically for ECMO patients. Because most published studies have used the older 3-level version, comparison with either general population index values or specific index values corresponding to ill individuals is limited. However, as 5 levels become the new paradigm in the HRQoL assessment, the use of EQ-5D-5L provides more accurate results, and thus, the benefits from its use outweigh the reported disadvantage.

## 5. Conclusions

Patients treated with VA ECMO for either CS or RCA have high in-hospital mortality, with RCA patients at higher risk of early death. However, once discharged from the hospital alive, most patients remain alive for years with a reasonable quality of life. Neurologic complications during initial hospital care decrease survivors’ ability to care for themselves, increase their pain and discomfort, and increase anxiety and depression.

## Figures and Tables

**Figure 1 jcm-11-06452-f001:**
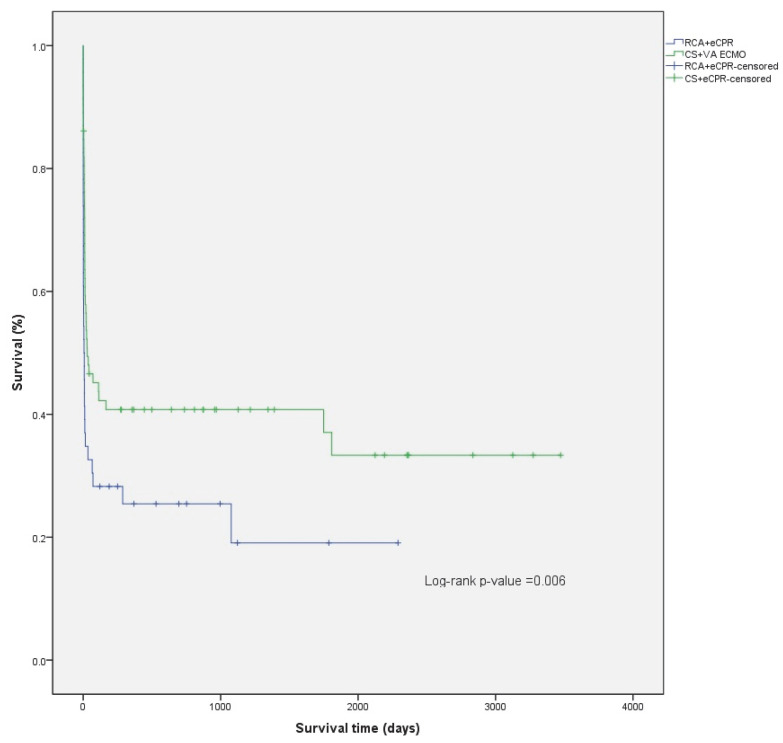
Comparison of Kaplan–Meier overall survival curves for CS and RCA patients.

**Figure 2 jcm-11-06452-f002:**
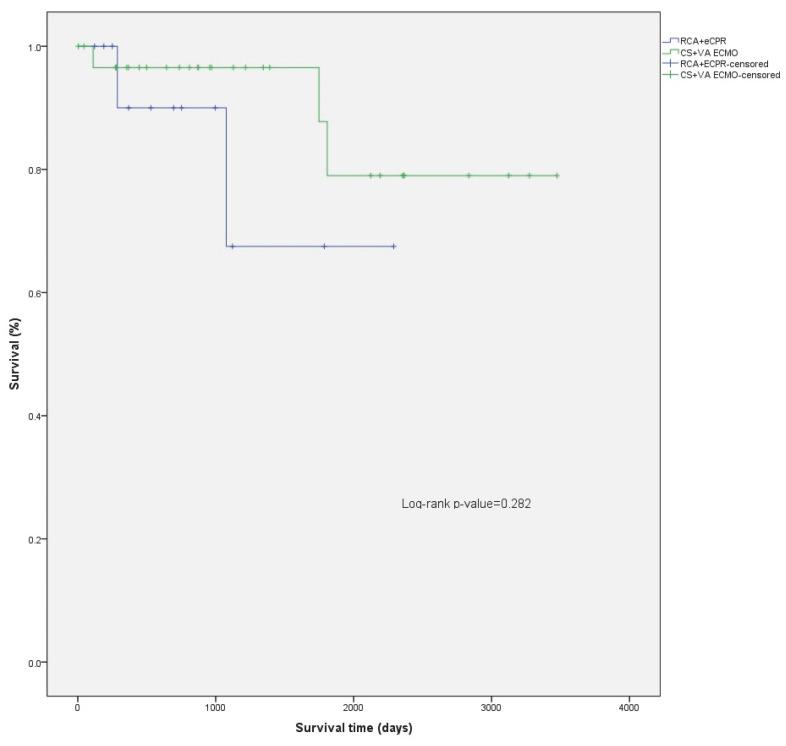
Comparison of survival after hospital discharge between CS and RCA patients.

**Figure 3 jcm-11-06452-f003:**
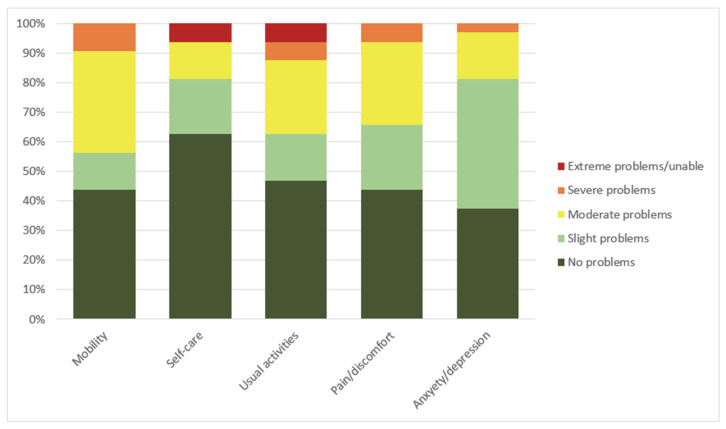
Frequency of severity of problems reported on the EQ-5D-5L (*n* = 32). Severity of problems represent 5 levels of health as defined by EQ-5D-5L descriptive system [24].

**Figure 4 jcm-11-06452-f004:**
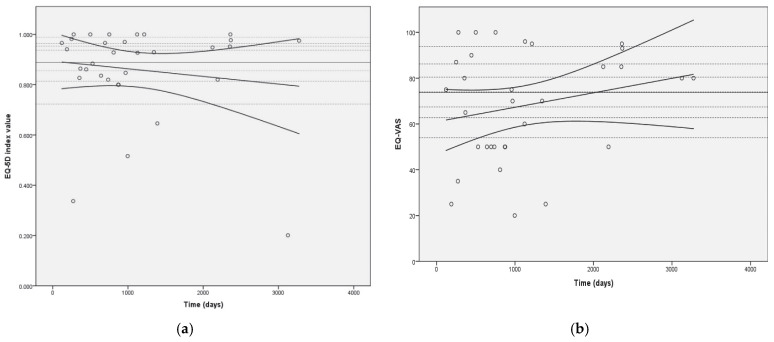
Distribution of EQ-5D-5L index values (**a**) and EQ-VAS (**b**) over time of follow-up. Legend: Horizontal dashed lines represent general population EQ-5D-5L index values by age groups for Figure 4a and general population EQ-VAS values by age group for Figure 4b (from top to bottom (EQ-5D-5L index value/EQ-VAS): 0.963/86.2—age 18–24 years; 0.953/83.9—age 25–34 years; 0.938/80.6—age 35–44 years; 0.898/74.0—age 45–54 years; 0.856/67.4—age 55–64 years; 0.813/62.8—age 65–74 years; and 0.723/54.0—age 75 and older; continuous horizontal line represents general population average of 0.888/73.7 [27].

**Table 1 jcm-11-06452-t001:** Characteristics of patients who participated in follow-up.

	All Participants	CS	RCA
	*n* = 32	*n* = 24	*n* = 8
Age (years), median (SD)	54.1 ± 8.8	52.2 ± 9.0	57.4 ± 7.7
Male sex, *n* (%)	24 (75.0%)	16 (66.7%)	8 (100.0%)
Severe pre-existing health problems ^1^	10 (31.3%)	10 (41.7%)	0 (0.0%)
Etiology			
Acute coronary syndrome	12 (37.5%)	6 (25.0%)	6 (75.0%)
Myocarditis	5 (15.6%)	5 (20.8%)	0 (0.0%)
Dilated cardiomyopathy	8 (25.0%)	8 (33.3%)	0 (0.0%)
Valvular heart disease	1 (3.1%)	1 (4.2%)	0 (0.0%)
Other	6 (18.8%)	4 (16.7%)	2 (25.0%)
Percutaneous cannulation	27 (84.4%)	19 (79.2%)	8 (100.0%)
Pre-cannulation cardiac arrest	17 (53.1%)	9 (37.5%)	8 (100.0%)
Duration of cardiac arrest (min)	35.2 ± 59.6	11.4 ± 3.8	80.4 ± 28.4
Lactate (mmol/L) ^2^	6.8 ± 4.4	6.9 ± 4.9	6.5 ± 2.6
SOFA	11.5 ± 3.5	11.0 ± 3.6	12.8 ± 2.7
APACHE II	22.7 ± 10.8	19.0 ± 9.8	32.2 ± 6.9
SAVE score	−5.6 ± 4.5	−5.0 ± 4.9	−7.1 ± 2.7
Renal replacement therapy ^3^	8 (25.0%)	7 (29.2%)	1 (12.5%)
Concomitant IABP	21 (65.6%)	16 (66.7%)	5 (62.5%)
Complications			
BARC 3-5 bleeding	7 (21.9%)	20 (83.3%)	5 (62.5%)
Thrombotic event	2 (6.3%)	2 (8.3%)	0 (0.0%)
Neurologic complications ^4^	2 (6.3%)	2 (8.3%)	0 (0.0%)
Cannulation related complications	1 (3.1%)	0 (0.0%)	1 (12.5%)
Limb ischemia	5 (15.6%)	4 (16.7%)	1 (12.5%)
Sepsis ^5^	1 (3.1%)	1 (4.2%)	0 (0.0%)
Heart transplant ^6^	5 (15.6%)	4 (16.7%)	1 (12.5%)
Long-term mechanical support	2 (6.3%)	2 (8.3%)	0 (0.0%)
Duration of ECMO support (days)	4.5 ± 4.5	5.0 ± 4.9	2.9 ± 2.4
ICU length of stay (days)	11.5 ± 7.3	11.3 ± 7.7	12.3 ± 6.6
Hospital length of stay (days)	51.6 ± 31.8	52.8 ± 35.4	48.1 ± 19.5

Abbreviations: APACHE II, Acute Physiology and Chronic Health Evaluation II; BARC, Bleeding Academic Research Consortium; CS, cardiogenic shock; RCA, refractory cardiac arrest treated with extracorporeal membrane oxygenation cardiopulmonary resuscitation; IABP, intra-aortic balloon pump; ICU, intensive care unit; SAVE, survival after veno-arterial ECMO; SD, standard deviation; SOFA, Sequential Organ Failure Assessment; ^1^, New York Heart Association Heart failure III or IV, history of stroke, or had organ transplant or organ failure requiring organ transplant; ^2^, first lactate value after ECMO initiation; ^3^, need of renal replacement therapy while on ECMO; ^4^, defined as stroke, intracranial bleeding, or proven brain death; ^5^, defined as positive blood cultures or proven source of infection with corresponding rise in laboratory markers of infection (leukocyte count, C reactive protein, and procalcitonin) and increased requirement of vasopressors; ^6^, heart transplant while on ECMO.

**Table 2 jcm-11-06452-t002:** Frequencies and proportions reported by dimension and level, EQ-VAS and EQ-5D-5L index values.

	All	CS	RCA	*p*
	*n* = 32	*n* = 24	*n* = 8	
Mobility				0.666
No problems	14 (43.8%)	10 (41.7%)	4 (50.0%)	
Slight problems	4 (12.5%)	4 (16.7%)	0 (0.0%)	
Moderate problems	11 (34.4%)	8 (33.3%)	3 (37.5%)	
Severe problems	3 (9.4%)	2 (8.3%)	1 (12.5%)	
Extreme problems/unable	0 (0.0%)	0 (0.0%)	0 (0.0%)	
Mean score ± SD	2.1 ± 1.1	2.1 ± 1.1	2.1 ± 1.2	0.934
Self-care				0.094
No problems	20 (62.5%)	12 (50%)	8 (100.0%)	
Slight problems	6 (18.8%)	6 (25.0%)	0 (0.0%)	
Moderate problems	4 (12.5%)	4 (16.7%)	0 (0.0%)	
Severe problems	0 (0.0%)	0 (0.0%)	0 (0.0%)	
Extreme problems/unable	2 (6.3%)	2 (8.3%)	0 (0.0%)	
Mean score ± SD	1.7 ± 1.1	1.9 ± 1.2	1.0 ± 0.0	0.001
Usual activities				0.209
No problems	15 (46.9%)	9 (37.5%)	6 (75.9%)	
Slight problems	5 (15.6%)	4 (16.7%)	1 (12.5%)	
Moderate problems	8 (25.0%)	8 (33.3%)	0 (0.0%)	
Severe problems	2 (6.3%)	1 (4.2%)	1 (12.5%)	
Extreme problems/unable	2 (6.3%)	2 (8.3%)	0 (0.0%)	
Mean score ± SD	2.1 ± 1.3	2.3 ± 1.3	1.5 ± 1.1	0.106
Pain/discomfort				0.206
No problems	14 (43.8%)	9 (37.5%)	5 (62.5%)	
Slight problems	7 (21.9%)	5 (25.8%)	2 (25.0%)	
Moderate problems	9 (28.1%)	9 (39.1%)	0 (0.0%)	
Severe problems	2 (6.3%)	1 (4.2%)	0 (0.0%)	
Extreme problems/unable	0 (0.0%)	0 (0.0%)	1 (12.5%)	
Mean score ± SD	2.0 ± 1.0	2.1 ± 1.0	1.6 ± 1.1	0.303
Anxiety/depression				0.165
No problems	12 (37.5%)	7 (29.2%)	5 (62.5%)	
Slight problems	14 (43.8%)	13 (54.2%)	1 (12.5%)	
Moderate problems	5 (15.6%)	3 (12.5%)	2 (25.0%)	
Severe problems	1 (3.1%)	1 (4.2%)	0 (0.0%)	
Extreme problems/unable	0 (0.0%)	0 (0.0%)	0 (0.0%)	
Mean score ± SD	1.8 ± 0.8	1.9 ± 0.8	1.6 ± 0.9	0.436
EQ-VAS	68.0 ± 24.2	71.2 ± 22.4	58.4 ± 28.2	0.198
Index value	0.864 ± 0.191	0.850 ± 0.201	0.907 ± 0.162	0.434

Abbreviation: CS, cardiogenic shock; CA, cardiac arrest treated with extracorporeal membrane oxygenation cardiopulmonary resuscitation; SD, standard deviation.

## Data Availability

The data presented in this study are available on request from the corresponding author. The data are not publicly available due to privacy.

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
