# Peer review of "Long-Term Survival and Quality of Life in Non-Surgical Adult Patients Supported with Veno-Arterial Extracorporeal Oxygenation"

_jcm, 2022, doi:10.3390/jcm11216452_

Round 1
Reviewer 1 Report
Line 184, mean follow up time and other values would be better to be expressed with median follow up with interquartile range.
Table 1, Survival at 30 day 1 year 3 years would be better be compared with log rank test rather than comparing them directly on each time points. This can be incorporated in Kaplan-Meier survical curves in Figure 1 and 2.
This manuscript included multiple different topics and analyses in a single manuscript. Some of them should be separated or sent to supplementary data.
Registry based data limits further interpretation of the unadjusted results. Risk adjustment is necessary to compare outcomes between cardiac arrest and cardiogenic shock group.
Author Response
We thank the reviewer for useful comments and the opportunity to improve the manuscript.
Reviewer comment:
Line 184, mean follow up time and other values would be better to be expressed with median follow up with interquartile range.
Authors replay:
Mean follow-up time was changed to median follow-up time and interquartile range as suggested. This change was also included in the statistical analysis section in Materials and Methods.
The manuscript was changed as following:
Statistical analisys
Follow-up time was calculated using the reverse Kaplan-Meier method and expressed as median and interquartile range (IQR).
Results
The median follow-up time was 969 days (IQR 500-2192) (2.7, IQR 1.4-6.0 years) for the patients studied, 1128 days (IQR 500-2355) (3.1, IQR 2.7 – 6.5 years) for the CS patients, and 752 days (IQR 369 – 1122) (2.1, IQR 1.01 – 3.1 years) for the eCPR patients.
Reviewer comment:
Table 1, Survival at 30 day 1 year 3 years would be better be compared with log rank test rather than comparing them directly on each time points. This can be incorporated in Kaplan-Meier survical curves in Figure 1 and 2.
Authors replay:
The comparison of survival curves was performed using the log-rank test, and the results are included in Figure 1 and Figure 2. As suggested in the next comment, it was decided to remove Table 1 from the main body of the manuscript and move it to the supplementary section. However, the comparison of survival at specific time points was left in the table to facilitate possible comparison.
The following changes were made:
Figure 1. Comparison of Kaplan-Meier overall survival curves for CS and RCA patients.
Figure 2. Comparison of survival after hospital discharge between CS and RCA patients.
Reviewer comment:
This manuscript included multiple different topics and analyses in a single manuscript. Some of them should be separated or sent to supplementary data.
Authors reply:
As suggested, portions of the manuscript have been moved to the supplementary section. Specifically, Table 1 and Table 4 were removed from the main body of the manuscript and submitted as supplementary material – Table A1 and Table A2.
Reviewer comment:
Registry based data limits further interpretation of the unadjusted results. Risk adjustment is necessary to compare outcomes between cardiac arrest and cardiogenic shock group.
Authors reply:
Preexisting risk factors were included in Table A1 for additional risk adjustment. To reduce potential confounding factors associated with the observational study design cox regression analysis was then performed using the following covariates: age, sex, arterial hypertension, diabetes, dyslipidemia, smoking, coronary artery disease, peripheral artery disease, BMI, and shock etiology. The adjusted survival was consistent with unadjusted analysis. Tables A3 and A4 and Figures A1 and A2 were added to appendix.
The following statement was added to Materials and Methods - Statistical Analysis:
Multivariate Cox regression analysis was then performed using as covariates the predictors of mortality, including age, sex, arterial hypertension, diabetes, dyslipidemia, smoking, coronary artery disease, peripheral artery disease, BMI, and shock etiology.
Figure A1. Cox Regression overall survival curves for CS and RCA patients.
Table A3. Cox Regression covariates analysis overall survival
|
|
P Value |
HR |
95,0% CI for HR Low High |
|
|
Arterial hypertension |
0.412 |
1.241 |
0.740 |
2.082 |
|
Diabetes |
0.264 |
0.708 |
0.386 |
1.299 |
|
Smoking |
0.067 |
0.609 |
0.359 |
1.035 |
|
Dyslipidemia |
0.128 |
0.616 |
0.331 |
1.149 |
|
Coronary artery disease |
<0.001 |
3.076 |
1.649 |
5.736 |
|
Peripheral artery disease |
0.085 |
2.433 |
0.885 |
6.686 |
|
Shock etiology |
0.072 |
1.149 |
0.988 |
1.337 |
|
Age |
0.951 |
1.001 |
0.976 |
1.026 |
|
Sex |
0.431 |
0.791 |
0.442 |
1.416 |
|
BMI |
<0.001 |
1.106 |
1.053 |
1.163 |
Abbreviations: CI – Confidential Interval, HR – Hazard Ratio
Figure A2. Cox Regression after hospital discharge between CS and RCA patients
Table A4. Cox Regression covariates analysis after hospital discharge
|
|
P Value |
HR |
95,0% CI For HR Low High |
|
|
Arterial Hypertension |
0.611 |
0.229 |
0.001 |
67.385 |
|
Diabetes |
0.709 |
1.964 |
0.056 |
68.292 |
|
Smoking |
0.862 |
1.333 |
0.053 |
33.698 |
|
Dyslipidemia |
0.626 |
2.089 |
0.108 |
40.274 |
|
Coronary artery disease |
0.657 |
0.400 |
0.007 |
22.885 |
|
Peripheral artery disease |
/ |
/ |
/ |
/ |
|
Shock etiology |
0.708 |
0.771 |
0.197 |
3.011 |
|
Age |
0.599 |
1.047 |
0.882 |
1.243 |
|
Sex |
0.461 |
0.242 |
0.006 |
10.569 |
|
BMI |
0.169 |
0.670 |
0.379 |
1.186 |
Abbreviations: CI – Confidential Interval, HR – Hazard Ratio, / - constant covariates
Reviewer 2 Report
Many thanks for this very interesting and well-conducted study. As a participant in an ethics committee in my hospital, these results are valuable.
I have only a few suggestions to improve ease of understanding for all target audiences:
The introduction deserves to be reworked, especially the first paragraph (lines 30 to 41) which is very abrupt. Try to better establish what the concepts of eCPR and VA ECMO are and how to move from one to the other, to arrive at a better organized and systematized paragraph.
You do not discuss the large dispersion of the EQ-VAS in Figure 3-b. This one is however very obvious when one compares it to the dispersion of the EQ-5D in the figure 3-a. Your article would be enriched if you could let us know what this fact inspires you. Furthermore, do you have the value of the EQ-VAS in the general population? and if so, could you include it in figure 3-b, as you do for the 5D-EQ in figure 3-a.
I spent a reasonable time reading your article, about 2 hours, and I did not perceive in the results the justification for one of your conclusions (line 385) "Neurological complications and major bleeding during initial care at hospital limit the survivors' long-term quality of life". Could you please argue this statement more clearly and make it more obvious.
Author Response
Reviewer comment: Comments and Suggestions for Authors
Many thanks for this very interesting and well-conducted study. As a participant in an ethics committee in my hospital, these results are valuable.
I have only a few suggestions to improve ease of understanding for all target audiences:
Authors replay:
We thank you for useful suggestions and the oportunity to improve the manuscript.
Reviewer comment: The introduction deserves to be reworked, especially the first paragraph (lines 30 to 41) which is very abrupt. Try to better establish what the concepts of eCPR and VA ECMO are and how to move from one to the other, to arrive at a better organized and systematized paragraph.
Authors reply:
Part of the introductory section has been expanded. Definitions of cardiogenic shock and ECPR were added with a brief overview of treatment. Additional references were added to this section (Richardson A, Resuscitation, 2017; Syed M, Cardiovascular revascularization med, 2022; Thiele H, Eur Heart J, 2019; Richardson A, ASAIO J, 2021; Kosmopoulos M, Hell J Cardiol, 2020).
Introduction
Veno-arterial extracorporeal membrane oxygenation (VA ECMO) is increasingly used for mechanical circulatory support of patients with either cardiogenic shock (CS) or refractory cardiac arrest (RCA) treated with extracorporeal membrane oxygenation cardiopulmonary resuscitation (eCPR) (1,2). Cardiogenic shock is defined as inability of the heart to maintain adequate cardiac output in accordance with metabolic demands attributed to predominantly cardiac pathology. When established treatment with fluid status optimization, vasopressors and inotropes, mechanical ventilation, and etiologic treatment of cardiogenic shock, whenever possible, fails to restore adequate cardiac output, temporary mechanical circulatory support with VA ECMO is an option to restore adequate organ perfusion (3). Randomized trials are underway to investigate whether mechanical support by extracorporeal membrane oxygenation (ECMO) provides a survival benefit in cardiogenic shock. Cardiac arrest is an extreme form of cardiogenic shock, and when conventional cardiopulmonary resuscitation fails to restore sustained heart function, cannulation while performing chest compressions and initiating VA ECMO, termed eCPR, is often the only hope for survival (4). We have the first randomized evidence that it is indeed superior to conventional cardiopulmonary resuscitation for cardiac arrest (5). Currently, the reported survival rate at hospital discharge for patients with cardiogenic shock supported with VA ECMO is 41-64% (6,7), whereas for refractory cardiac arrest patients treated with eCPR, it is highly dependent on the site of cardiac arrest (in-hospital or out-of-hospital) (8–10) and the site of cannulation (cath lab or out-of-hospital) (9,11). Currently, the survival rate for RCA treated with eCPR ranges from 8-54% (1,8,9,12).
There is a body of evidence supporting the short-term survival benefit of VA ECMO for CS and RCA (13–16). However, little is known about long-term survival beyond the first year post-discharge and the health-related quality of life (HRQoL) in discharged patients.
Cardiac arrest is just extreme form of CS and it is often difficult to distinguish between CS being cannulated for VA ECMO and RCA patients who require eCPR. Because of the different baseline characteristics according to treatment strategy (17) these two groups were evaluated separately.
Patients treated for CS or RCA with VA ECMO belong to a very sick population with high in-hospital mortality and complication rates due to both the severity of the underlying disease and the ECMO treatment. Many of these problems may persist after discharge and affect post-hospital recovery and quality of life.
The EuroQol - 5 dimensional (EQ-5D) and the short-form health survey with 36 questions (SF-36) are the most commonly used questionnaires to assess health-related quality of life in clinical trials. The EQ-5D questionnaire exists in two forms: 3-level (EQ-5D-3L) and 5-level (EQ-5D-5L). The EQ-5D-5L is a newer version, available since 2011, that provides more precise and valid results and is recommended as the assessment tool of choice (18); therefore, it was used in our study.
This report aims to assess the long-term survival of patients treated with VA ECMO and their HRQoL, to compare differences in long-term survival and HRQoL between patients with cardiogenic shock and patients with refractory cardiac arrest, and to investigate their possible influences on HRQoL.
Reviewer comment:
You do not discuss the large dispersion of the EQ-VAS in Figure 3-b. This one is however very obvious when one compares it to the dispersion of the EQ-5D in the figure 3-a. Your article would be enriched if you could let us know what this fact inspires you. Furthermore, do you have the value of the EQ-VAS in the general population? and if so, could you include it in figure 3-b, as you do for the 5D-EQ in figure 3-a.
Authors replay:
As with the population index values in Figure 3a, the population values for age groups and the population average value EQ-VAS have been added to Figure 3b
A paragraph has been added to the Discussion section to explain and comment on wider dispersity of EQ-VAS.
Hellevuo et al. (40) indicate that HRQoL remains good in survivors 6 months after cardiac arrest and that it is mainly influenced by HRQoL before cardiac arrest. To some extent, this might be true for VA ECMO patients. RCA patients in our cohort were mostly healthy before cardiac arrest, in contrast to CS patients who often had severe preexisting health problems. The same trend was seen in EQ-5D-5L index values, which were numerically higher in RCA patients and had lower numerical EQ-VAS scores compared to CS patients. EQ-VAS represents a more personal perception of health, whereas index values are used to represent socially perceived health status. EQ-VAS, which has been reported in other published series, was quite similar to ours (41). A higher dispersion is observed on EQ-VAS, especially in the RCA group. There could be several explanations for the higher dispersion. The EQ-VAS also has greater dispersion in the general population (42). In contrast to the EQ -5D-5L index values, which decrease over time, we observed a trend toward improvement in EQ-VAS in our population. Because patients in the RCA group had a shorter follow-up period, one might expect an improvement in personal health perception over time and also a smaller dispersion of EQ-VAS.
Figure 3. Distribution of EQ-5D-5L index values (a) and EQ-VAS (b) over time of follow-up.
Legend: Horizontal dashed lines represent general population EQ-5D-5L index values by age groups for figure 3a and general population EQ-VAS values by age group for figure 3b (from top to bottom (EQ-5D-5L index value / EQ-VAS): 0.963 / 86.2 – age 18-24 years; 0.953 / 83.9 – age 25-34 years; 0.938 / 80.6 – age 35-44 years; 0.898 / 74.0 – age 45-54 years; 0.856 / 67.4 – age 55-64 years; 0.813 / 62.8 – age 65-74 years; 0.723 / 54.0 – age 75 and older; continuous horizontal line represents general population average of 0.888 / 73.7 (38).
Reviewer comment: I spent a reasonable time reading your article, about 2 hours, and I did not perceive in the results the justification for one of your conclusions (line 385) "Neurological complications and major bleeding during initial care at hospital limit the survivors' long-term quality of life". Could you please argue this statement more clearly and make it more obvious.
Authors replay:
We thank the reviewer for pointing out the inaccurate conclusion.
We have reworded the last sentence of the conclusion to make it more precise. As stated by the reviewer, neurological complications affected only some dimensions of EQ -5D but not the total score.
Conclusions
Patients treated with VA ECMO for either CS or RCA have high in-hospital mortality with RCA patients at higher risk of early death. However, once discharged from the hospital alive, most patients remain alive for years with a reasonable quality of life. Neurologic complications during initial hospital care decrease survivors’ ability to care for themselves, increase their pain and discomfort, and increase anxiety and depression.
Reviewer 3 Report
This manuscript raises an important issue and provides relevant results. It is scientifically sound and appeals to a broad readership. But some remarks still remain. 1. Citations in a scientific article must be accurate. This cannot be said about the first quote [1]. In the cited article, when using ECMO, there is no division into “patients with cardiogenic shock (CS) or refractory cardiac arrest (RAA)” (lines 31,32) [1]. The cited article talks about ECMO more generally. Thus, for adults, this one focuses on the use of ECMO for cardiac or pulmonary dysfunction, and not for various types of heart fail, as in the present article. The citation should be adjusted according to the content of the cited article. 2. The division of all patients into these two groups, with CS and CA who required eCPR, is rather ambiguous. It is known that CS and CA frequently occur together, and CS can cause CA. The authors divided these two similar conditions according to the presence of CPR. Maybe the authors will tell if they experienced difficulties in identifying these groups and will provide literature data on this topic. 3. There are two groups in this study: СS and СA. One has CPR, the other does not. Therefore, the second group is called CPR. But only the second CPR group has “e” in front (which means ECMO). It might be better to call this group CA + CPR, since the first group is also "e", because both groups used ECMO. 4. Table 1: missing unit of Lactate (mmol/l); Table 2: missing unit in " Duration of cardiac arrest” (min?). 5. The links in the article are not quite correct. They should appear at the end of the quoted sentence before a dot. In conclusion, in spite of these minor comments, the article may be recommended for publication.Author Response
This manuscript raises an important issue and provides relevant results. It is scientifically sound and appeals to a broad readership. But some remarks still remain.
Reviewer comment:
- Citations in a scientific article must be accurate. This cannot be said about the first quote [1]. In the cited article, when using ECMO, there is no division into “patients with cardiogenic shock (CS) or refractory cardiac arrest (RAA)” (lines 31,32) [1]. The cited article talks about ECMO more generally. Thus, for adults, this one focuses on the use of ECMO for cardiac or pulmonary dysfunction, and not for various types of heart fail, as in the present article. The citation should be adjusted according to the content of the cited article.
Authors replay:
We thank the reviewer for pointing out the inaccurate citation. The citation has been changed to adequately support the claim made in the first sentence of the introduction. One citation was removed and two others added (Richardson A, Resuscitation, 2022; Syed M, Cardiovasc Revascularization Med, 2022). Other references were also reviewed.
Reviewer comment:
- The division of all patients into these two groups, with CS and CA who required eCPR, is rather ambiguous. It is known that CS and CA frequently occur together, and CS can cause CA. The authors divided these two similar conditions according to the presence of CPR. Maybe the authors will tell if they experienced difficulties in identifying these groups and will provide literature data on this topic.
Replay:
Despite clear definitions of CA and eCPR, it is often difficult to distinguish between the two groups. In the introduction to the manuscript, an additional paragraph on this topic and a reference (Kosmopoulos M, Hell J Cardiol, 2020) were added to support the claim that the 2 groups differ in baseline characteristics based on treatment strategy to support the separate assessment.
Cardiac arrest is just extreme form of CS and it is often difficult to distinguish between CS being cannulated for VA ECMO and RCA patients who require eCPR. Because of the different baseline characteristics according to treatment strategy (17) these two groups were evaluated separately.
Reviewer comment:
- There are two groups in this study: СS and СA. One has CPR, the other does not. Therefore, the second group is called CPR. But only the second CPR group has “e” in front (which means ECMO). It might be better to call this group CA + CPR, since the first group is also "e", because both groups used ECMO.
Authors reply:
To focus on the main indication for VA ECMO treatment and to avoid confusion, the eCPR group has been renamed refractory cardiac arrest (RCA) throughout the manuscript.
Reviewer comment: 4. Table 1: missing unit of Lactate (mmol/l); Table 2: missing unit in " Duration of cardiac arrest” (min?).
Authors replay:
The missing units for lactate were added to Table 1 and Table 2, as was the unit for the duration of cardiac arrest in minutes in Table 2. However, as suggested by reviewer 1 to narrow the subject of the manuscript, Table 1 and Table 4 were removed from the main part of the manuscript and included in the supplemental section.
Table A1. Patient characteristics and comparison between CS and CA patients.
|
|
All patients |
CS |
RCA |
P |
|
|
n=118 |
n=72 |
n=46 |
|
|
Age (years), mean (SD) |
53.4±11.5 |
53.9±11.3 |
52.7±12.0 |
0.613 |
|
Male sex, n (%) |
93 (78.8%) |
54 (75.0%) |
39(84.8%) |
0.205 |
|
Serious pre-existing health problems 1 |
31 (26.3%) |
24 (33.3%) |
7 (15.2%) |
0.029 |
|
Pre-existing risk factors |
|
|
|
|
|
Arterial hipertension |
46 (39.0%) |
24 (33.3%) |
22 (47.8%) |
0.115 |
|
Diabetes |
23 (19.5%) |
16 (22.2%) |
7 (15.2%) |
0.349 |
|
Dyslipidemia |
26 (22.0%) |
12 (16.7%) |
14 (30.4%) |
0.078 |
|
Smoking |
37 (31.4%) |
14 (19.4%) |
23 (50.0%) |
<0.001 |
|
Coronary artery disease |
52 (44.1%) |
41 (56.9%) |
11 (23.9%) |
<0.001 |
|
Peripheral artery disease |
7 (5.9%) |
4 (5.6%) |
3 (6.5%) |
0.828 |
|
Etiology |
|
|
|
0.015 |
|
Acute coronary syndrome |
60 (50.8%) |
33 (45.8%) |
27 (58.7%) |
|
|
Myocarditis |
12 (10.2%) |
11 (15.3%) |
1 (2.2%) |
|
|
Dilated cardiomyopathy |
19 (16.1%) |
16 (22.2%) |
3 (6.5%) |
|
|
Pulmonary embolism |
4 (3.4%) |
1 (1.4%) |
3 (6.5%) |
|
|
Valvular heart disease |
3 (2.5%) |
1 (1.4%) |
2 (4.3%) |
|
|
Other |
20 (16.9%) |
10 (13.9%) |
10 (21.7%) |
|
|
Percutaneous cannulation |
100 (84.7%) |
54 (75.0%) |
46 (100.0%) |
<0.001 |
|
Pre-cannulation cardiac arrest |
72 (61.0%) |
26 (36.1%) |
46 (100.0%) |
<0.001 |
|
Duration of cardiac arrest, mean (min) |
44±41 |
15±3 |
42±6 |
<0.001 |
|
Lactate (mmol/L)2 |
9.0±6.5 |
6.6±5.6 |
13.1±6.0 |
<0.001 |
|
SOFA |
11.9±4.1 |
10.9±4.0 |
13.9±3.5 |
<0.001 |
|
APACHE II |
24.7±10.7 |
19.8±9.3 |
33.7±6.5 |
<0.001 |
|
SAVE score |
-5.7±4.9 |
-4.6±5.6 |
-7.3±3.1 |
0.001 |
|
Renal replacement therapy 3 |
34 (28.8%) |
25 (34.7%) |
9 (19.6%) |
0.107 |
|
Concomitant IABP |
68 (57.6%) |
49 (68.1%) |
19 (41.3%) |
0.004 |
|
Complications |
|
|
|
|
|
Bleeding (BARC 3-5) |
50 (42.4%) |
29 (40.3%) |
25 (54.3%) |
0.564 |
|
Thrombotic event |
11 (9.3%) |
9 (12.5%) |
2 (4.3%) |
0.159 |
|
Neurologic complications 4 |
16 (13.6%) |
9 (12.5%) |
7 (15.2%) |
0.407 |
|
Cannulation related complications |
10 (8.5%) |
4 (5.6%) |
6 (13.0%) |
0.154 |
|
Limb ischemia |
32 (27.1%) |
19 (26.4%) |
13 (28.3%) |
0.435 |
|
Sepsis 5 |
17 (14.4%) |
13 (18.1%) |
4 (8.7%) |
0.245 |
|
Unable to start ECMO |
3 (2.5%) |
2 (2.8%) |
1 (2.2%) |
0.839 |
|
Duration of ECMO support (days) |
5.5±5.8 |
6.6±5.5 |
3.9±6.1 |
0.021 |
|
Heart transplant 6 |
10 (8.5%) |
9 (12.5%) |
1 (2.2%) |
0.049 |
|
Long term mechanical support |
9 (7.6%) |
7 (9.7%) |
2 (4.3%) |
0.283 |
|
ICU length of stay (days) |
8.3±8.8 |
10.1±9.4 |
5.5±6.9 |
0.003 |
|
Hospital length of stay (days) |
27.9±32.7 |
33.3±35.7 |
20.3±26.3 |
0.028 |
|
Survival to decannulation |
51 (43.2%) |
32 (44.4%) |
19 (41.3%) |
0.442 |
|
Survival at hospital discharge |
44 (37.3%) |
31 (43.1%) |
13 (28.3%) |
0.105 |
|
Survival at 30 days |
51 (43.2%) |
35 (48.6%) |
16 (34.8%) |
0.139 |
|
|
N=101 |
N=64 |
N=37 |
|
|
Survival at 1 year 7 |
34 (33.7%) |
25 (39.1%) |
9 (24.3%) |
0.131 |
|
|
N=71 |
N=50 |
N=21 |
|
|
Survival at 3 years 7 |
18 (25.4%) |
15 (30.0%) |
3 (14.3%) |
0.168 |
Abbreviations: APACHE II - Acute Physiology and Chronic Health Evaluation II, BARC - Bleeding Academic Research Consortium, CS – cardiogenic shock, RCA – refractory cardiac arrest treated with extracorporeal membrane oxygenation cardiopulmonary resuscitation, IABP – Intra-aortic balloon pump, ICU – intensive care unit, SAVE - Survival after Veno-Arterial ECMO, SD – standard deviation, SOFA - Sequential Organ Failure Assessment; 1 - New York heart Association Heart failure III or IV, history of stroke, had organ transplant or organ failure requiring organ transplant; 2 – first lactate value after ECMO initiation; 3 – need of renal replacement therapy while on ECMO; 4 – defined as stroke, intracranial bleeding or proven brain death; 5 – defined as (positive blood cultures or proven source of infection with corresponding rise in laboratory markers of infection (leukocyte count, C reactive protein and procalcitonin) and increased requirement of vasopressors; 6 – heart transplant while on ECMO; 7 – calculated only for patients not censored.
Table 1. Characteristics of patients who participated in follow-up.
|
|
All participants |
CS |
RCA |
|
|
n=32 |
n=24 |
n=8 |
|
Age (years), median (SD) |
54.1±8.8 |
52.2±9.0 |
57.4±7.7 |
|
Male sex, n (%) |
24 (75.0%) |
16 (66.7%) |
8 (100.0%) |
|
Severe pre-existing health problems 1 |
10 (31.3%) |
10 (41.7%) |
0 (0.0%) |
|
Etiology |
|
|
|
|
Acute coronary syndrome |
12 (37.5%) |
6 (25.0%) |
6 (75.0%) |
|
Myocarditis |
5 (15.6%) |
5 (20.8%) |
0 (0.0%) |
|
Dilated cardiomyopathy |
8 (25.0%) |
8 (33.3%) |
0 (0.0%) |
|
Valvular heart disease |
1 (3.1%) |
1 (4.2%) |
0 (0.0%) |
|
Other |
6 (18.8%) |
4 (16.7%) |
2 (25.0%) |
|
Percutaneous cannulation |
27 (84.4%) |
19 (79.2%) |
8 (100.0%) |
|
Pre-cannulation cardiac arrest |
17 (53.1%) |
9 (37.5%) |
8 (100.0%) |
|
Duration of cardiac arrest (min) |
35.2±59.6 |
11.4±3.8 |
80.4±28.4 |
|
Lactate (mmol/L)2 |
6.8±4.4 |
6.9±4.9 |
6.5±2.6 |
|
SOFA |
11.5±3.5 |
11.0±3.6 |
12.8±2.7 |
|
APACHE II |
22.7±10.8 |
19.0±9.8 |
32.2±6.9 |
|
SAVE score |
-5.6±4.5 |
-5.0±4.9 |
-7.1±2.7 |
|
Renal replacement therapy 3 |
8 (25.0%) |
7 (29.2%) |
1 (12.5%) |
|
Concomitant IABP |
21 (65.6%) |
16 (66.7%) |
5 (62.5%) |
|
Complications |
|
|
|
|
BARC 3-5 bleeding |
7 (21.9%) |
20 (83.3%) |
5 (62.5%) |
|
Thrombotic event |
2 (6,3%) |
2 (8.3%) |
0 (0.0%) |
|
Neurologic complications 4 |
2 (6.3%) |
2 (8.3%) |
0 (0.0%) |
|
Cannulation related complications |
1 (3.1%) |
0 (0.0%) |
1 (12.5%) |
|
Limb ischemia |
5 (15.6%) |
4 (16.7%) |
1 (12.5%) |
|
Sepsis 5 |
1 (3.1%) |
1 (4.2%) |
0 (0.0%) |
|
Heart transplant 6 |
5 (15.6%) |
4 (16.7%) |
1 (12.5%) |
|
Long term mechanical support |
2 (6.3%) |
2 (8.3%) |
0 (0.0%) |
|
Duration of ECMO support (days) |
4.5±4.5 |
5.0±4.9 |
2.9±2.4 |
|
ICU length of stay (days) |
11.5±7.3 |
11.3±7.7 |
12.3±6.6 |
|
Hospital length of stay (days) |
51.6±31.8 |
52.8±35.4 |
48.1±19.5 |
Abbreviations: APACHE II - Acute Physiology and Chronic Health Evaluation II, BARC - Bleeding Academic Research Consortium, CS – cardiogenic shock, RCA – refractory cardiac arrest treated with extracorporeal membrane oxygenation cardiopulmonary resuscitation, IABP – Intra-aortic balloon pump, ICU – intensive care unit, SAVE - Survival after Veno-Arterial ECMO, SD – standard deviation, SOFA - Sequential Organ Failure Assessment; 1 - New York heart Association Heart failure III or IV, history of stroke, had organ transplant or organ failure requiring organ transplant; 2 – first lactate value after ECMO initiation; 3 – need of renal replacement therapy while on ECMO; 4 – defined as stroke, intracranial bleeding or proven brain death; 5 – defined as (positive blood cultures or proven source of infection with corresponding rise in laboratory markers of infection (leukocyte count, C reactive protein and procalcitonin) and increased requirement of vasopressors; 6 – heart transplant while on ECMO.
Reviewer: 5. The links in the article are not quite correct. They should appear at the end of the quoted sentence before a dot.
Reply:
Citations have been corrected to be consistent throughout the manuscript and placed before a period.
Reviewer comment: In conclusion, in spite of these minor comments, the article may be recommended for publication.
Thank you!
Round 2
Reviewer 1 Report
Thank you for addressing the comments successfully. Continue to edit errata. I have no other comments to add.